# Study protocol for the 'HelpMeDoIt!' randomised controlled feasibility trial: an app, web and social support-based weight loss intervention for adults with obesity

Lynsay Matthews,[1] Juliana Pugmire,[1] Laurence Moore,[1] Mark Kelson,[2] Alex McConnachie,[3] Emma McIntosh,[4] Sarah Morgan-Trimmer,[5] Simon Murphy,[6] Kathryn Hughes,[7] Elinor Coulman,[8] Olga Utkina-Macaskill,[1] Sharon Anne Simpson[1]

For numbered affiliations see end of article.

**Correspondence to**
Dr Lynsay Matthews;
Lynsay.Matthews@glasgow.ac.uk

## ABSTRACT

**Introduction** HelpMeDoIt! will test the feasibility of an innovative weight loss intervention using a smartphone app and website. Goal setting, self-monitoring and social support are three key facilitators of behaviour change. HelpMeDoIt! incorporates these features and encourages participants to invite 'helpers' from their social circle to help them achieve their goal(s).

**Aim** To test the feasibility of the intervention in supporting adults with obesity to achieve weight loss goals.

**Methods and analysis** 12-month feasibility randomised controlled trial and accompanying process evaluation. Participants (n=120) will be adults interested in losing weight, body mass index (BMI)≥30 kg/m$^2$ and smartphone users. The intervention group will use the app/website for 12 months. Participants will nominate one or more helpers to support them. Helpers have access to the app/website. The control group will receive a leaflet on healthy lifestyle and will have access to HelpMeDoIt! after follow-up. The key outcome of the study is whether prespecified progression criteria have been met in order to progress to a larger randomised controlled effectiveness trial. Data will be collected at baseline, 6 and 12 months. Outcomes focus on exploring the feasibility of delivering the intervention and include: (i) assessing three primary outcomes (BMI, physical activity and diet); (ii) secondary outcomes of waist/hip circumference, health-related quality of life, social support, self-efficacy, motivation and mental health; (iii) recruitment and retention; (iv) National Health Service (NHS) resource use and participant borne costs; (v) usability and acceptability of the app/website; and (vi) qualitative interviews with up to 50 participants and 20 helpers on their experiences of the intervention. Statistical analyses will focus on feasibility outcomes and provide initial estimates of intervention effects. Thematic analysis of qualitative interviews will assess implementation, acceptability, mechanisms of effect and contextual factors influencing the intervention.

**Ethics and dissemination** The protocol has been approved by the West of Scotland NHS Research Ethics Committee (Ref: 15/WS/0288) and the University of Glasgow MVLS College Ethics Committee (Ref:

## Strengths and limitations of this study

► Importance of the study: HelpMeDoIt! will test the feasibility of an innovative weight loss intervention using a smartphone app and website that seeks to engage helpers drawn from an individual's social network to support them to achieve their goals. This work is a critical step to inform the value and design of a potential effectiveness trial. If the intervention is found to be effective in a subsequent full trial, it has the potential to reach a large number of people at a low cost.

► Robust intervention development: The study team has used a collaborative person-centred approach to ensure that the resulting intervention is based on insights from a range of potential users.

► Theory-based intervention: The HelpMeDoIt! intervention incorporates evidence-based behaviour change theory. A key innovation of the app is that it is not only used by the individual but aims to mobilise social support.

► Data collection methods: The study uses a mixed-methods approach (anthropometrics, interviews, questionnaires and web/app analytics) to address the research questions.

► Generalisability: This study will be undertaken in Glasgow, Scotland. Although this may limit generalisability, the feasibility findings will inform the development of a larger effectiveness trial.

200140108). Findings will be disseminated widely through peer-reviewed publication and conference presentations.

**Trial registration number** ISRCTN85615983.

## INTRODUCTION

Poor diet, physical inactivity and high body mass index (BMI) have been highlighted in the top 10 risk factors for global burden of disease.[1] Preventative interventions, which are accessible, engaging and which

successfully improve health behaviours, are necessary to reverse current trends. Interventions to date have had limited impact and approaches which are known to work are not always adopted.[2] Novel interventions which incorporate effective approaches are therefore needed.

Technology offers opportunities to develop interventions that can reach a large proportion of the population at a low cost. In particular, smartphone apps and website-based interventions can be effective in influencing behaviour and reaching large numbers of people.[3–5] In 2015, internet access was available in 86% of UK households and accessed by 78% of adults either every day, or almost every day.[6] Smartphones were owned by 76% of adults, of whom >50% reported checking their phone within 5 min of waking.[7] Interventions delivered via these technologies also have the potential to reach people from lower socioeconomic groups, with 75% of people living in Scotland's 20% most deprived areas having access to the internet.[8] There is also evidence that these technologies can be effective with both younger and older people.[9 10]

Previous research has highlighted three key features important for behaviour change: (i) goal setting; (ii) self-monitoring; and (iii) social support.[11–13] The role of social support from family and friends is known to be particularly important in helping people to both achieve and *sustain* health behaviour change.[14 15] There are many smartphone apps (and accompanying websites) available that incorporate some of these key features (eg, 'Stickk',[16] 'MyFitnessPal').[17] However, a systematic review of the most popular apps for weight loss (n=28) found the majority were of inadequate quality, lacked evidence-based information on weight loss and lacked appropriate behaviour change techniques.[18]

Some apps provide an element of social support, such as the provision of a chat forum.[17] There is modest evidence to suggest that online social networks can positively impact health behaviour change.[19] However, online users are typically not known to each other and the apps are not designed to harness the 'offline world' support of family and friends from an individual's social network. Evidence indicates that support from key individuals in a person's life is more effective than that provided by anonymous online contacts.[20]

While the intervention elements of goal setting, monitoring and social support are well established and new technologies have shown promise, the evidence base is limited and theoretically underdeveloped.[9 21] Studies are often limited by small, short-term effects[22] and high attrition.[23 24] There are significant gaps in understanding how these elements work together, for example, how social support operates through personal networks mediated by new technologies, and what impact this has on mechanisms such as monitoring. There is a need to further explore their application and mechanisms of action.

In particular, social support and its relation to health behaviour change is undertheorised. It is not clear which type(s) of social support might be most effective for health behaviour change or how that support

should be promoted. There are different types of social support[13 14] and different kinds of support giving/receiving behaviours.[25] Social support can be conceptualised in varied ways in terms of who provides the support. Family, friends, influential people within existing social networks and fellow members of groups with a shared behavioural goal have been found to be effective in numerous behaviour change studies. These include increasing diet and exercise self-care for people with diabetes,[26] increasing weight loss in adults with obesity,[27] decreasing risk of HIV infection in men[28] and increasing psychological well-being in various populations.[29] The HelpMeDoIt! intervention aims to address the gaps mentioned by exploring the feasibility of an app which incorporates: (i) appropriate behaviour change techniques; (ii) evidence-based information on weight loss; and (iii) delivers this information via a platform which is both usable and acceptable for participants.

### Development of the HelpMeDoIt! intervention (stage 1)

We propose to test the feasibility of the HelpMeDoIt! intervention, an app-based and web-based resource promoting health behaviour change via three key features: goal setting, monitoring and social support. We used evidence from existing systematic reviews and relevant theory in developing the logic model (figure 1) which we then used to develop prototypes of the website and app. The intervention was developed iteratively and collaboratively over a 12-month period and involved the research team working closely with user representatives (n=36) and a software company. This early development work followed the Medical Research Council (MRC) guidance for the development and evaluation of complex interventions.[30] Full details of our intervention development will be published separately. In brief, we used formal development methods, including the: (i) 6SQUID approach (six steps in quality intervention development);[31] (ii) person-centred approach;[32] (iii) behavioural intervention technology (BIT) model;[33] and (iv) ongoing refinement of our intervention logic model and programme theory (figure 1). This helped to identify needs, targets and processes of change, in addition to possible barriers, facilitators and contextual factors that influence people's ability to perform the target behaviours.

### Aim

To test the feasibility and acceptability of the HelpMeDoIt! intervention in supporting adults with obesity to achieve weight loss goals and to identify the value and optimal design of a potential future effectiveness trial. Exploratory trials of this nature are a necessary first step in developing public health improvement interventions,[34] particularly where innovations and mechanisms, such as social support, are not well understood.

### METHODS AND ANALYSIS (STAGE 2)

The HelpMeDoIt! study has two stages (figure 2). Stage 1, outlined above, focused on the development and formative evaluation of the intervention. Stage 2, described

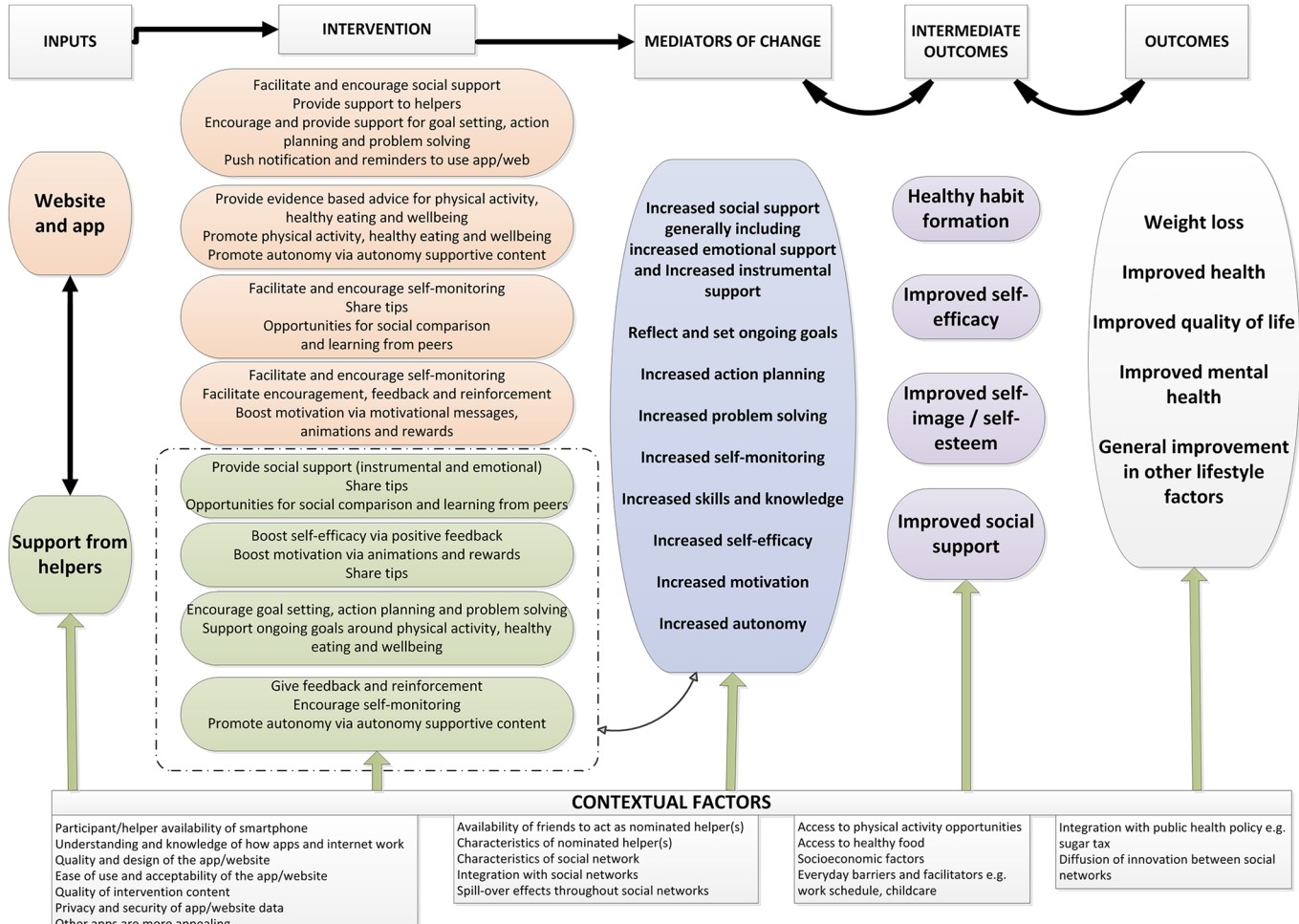

**Figure 1** The HelpMeDoIt! logic model.

here, focuses on implementing the intervention within a feasibility randomised controlled trial including process, outcome and health economic evaluation. The following methods adhere to the Standard Protocol Items Recommendations for Interventional Trials (SPIRIT) guidelines for the reporting of study protocols (see online supplementary appendix 1).[35 36]

### Study design and setting

HelpMeDoIt! is a feasibility randomised controlled trial conducted with adults with obesity living in Glasgow, Scotland (April 2016–February 2018).

### Participants

Participants are eligible for the trial if they meet the following inclusion/exclusion criteria.

### Inclusion criteria

▶ adults aged 18–70 years
▶ BMI of ≥30 kg/m$^2$
▶ trying to lose weight
▶ access to a smartphone and the internet.

### Exclusion criteria

▶ terminal illness

▶ previous bariatric surgery
▶ dementia
▶ pregnancy
▶ poor competence in English (resulting in inability to complete study materials)
▶ contraindications to physical activity
▶ previously a participant in stage 1 intervention development
▶ already being a nominated helper in the trial.

We will assess contraindications to physical activity using an adapted Physical Activity Readiness Questionnaire.[37] Anyone with a medical condition or taking medication or who thinks they may have a contraindication to physical activity will be advised to check with their own general practitioner (GP) before commencing any physical activity. We will ask women of childbearing age to let the study team know if they become pregnant at any point during the trial. Once recruited, pregnant women will not be excluded from the study as the intervention may still help them make healthy lifestyle choices. Also in a future trial the analysis would be intention to treat and we would not exclude women who become pregnant. They will be given a leaflet on diet and safe physical activity during pregnancy.

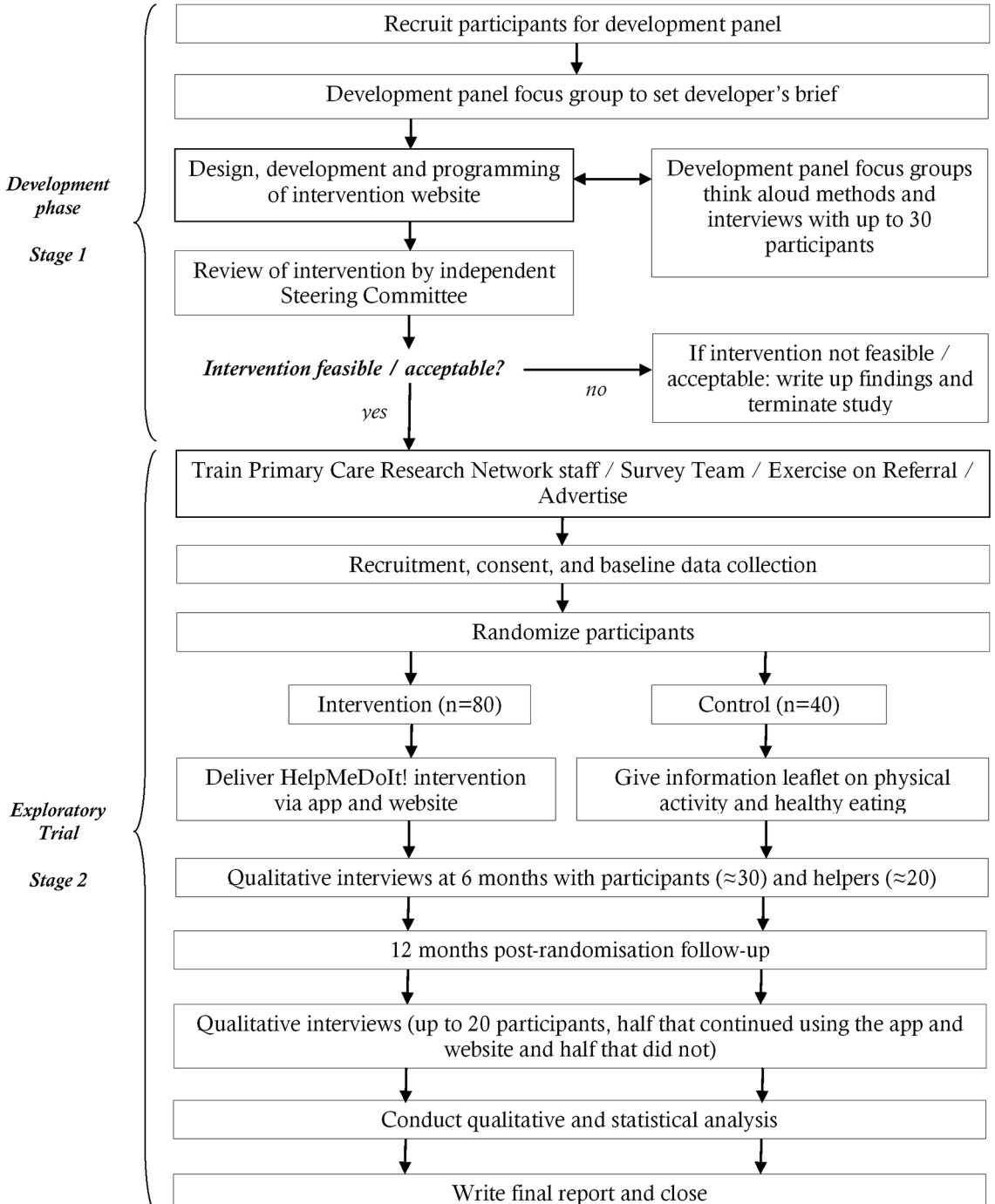

**Figure 2** HelpMeDoIt! study flow chart.

## Procedures

### Recruitment
A multipoint recruitment strategy will be employed to target a broad range of participants (eg, age, gender, socioeconomic status). This will primarily involve recruiting via: (i) online sources; (ii) primary care; and (iii) community sources.

### Online recruitment
Regular adverts will be placed on the Glasgow hub of Gumtree (a free online community advertising website). We will also establish a HelpMeDoIt! Facebook page

and Twitter account for posting up-to-date information. Interested individuals will be encouraged to express their interest to the study team, who will then send them a detailed participant information sheet.

### Primary care recruitment
We will collaborate with the Scottish Primary Care Research Network (SPCRN) to identify potential participants from GP records. SPCRN staff will liaise with GP practices on behalf of the study team and search patient databases for eligible participants based on the inclusion/exclusion criteria. Practice staff will exclude

vulnerable participants based on other known information. The SPCRN will post a recruitment pack (which includes a cover letter from their GP, study information sheet, contact details form and a prepaid envelope) to the agreed list of patients. Interested participants are asked to complete the contact details form and return the form in the envelope to the study team.

## Community recruitment

We will advertise via local press, slimming clubs, weight management clinics, exercise on referral service and by exhibiting study posters in multiple community locations. In addition to the strategies already mentioned, we aim to target as many males as possible via our local knowledge of community venues (eg, barbers, local football clubs) as recruiting men onto weight loss trials is known to be challenging.

All individuals who express an interest in the study will receive a participant information sheet and have at least 1 week to consider taking part in the study. A trained fieldworker will contact individuals by telephone to check eligibility and to arrange an appointment for informed consent and baseline data collection. Fieldworkers will meet with participants at a place of their choice, which could include their home (in which case our lone working policy will be followed) or a room at our research unit.

## Randomisation

We are most interested in exploring the feasibility of the intervention and so will randomise in a 2:1 ratio into intervention and control. Of 120 participants, approximately 80 participants will be allocated to the intervention group and 40 to the control group. Participants will be allocated using a mixed randomisation/minimisation algorithm to ensure balance with respect to gender and BMI ($<40, \geq 40 \, \text{kg/m}^2$). In blocks of 15 participants, 12 will be assigned according to the minimisation algorithm (designed to maintain as close to a 2:1 allocation ratio within strata defined by each minimisation factor) and 3 will be allocated (in a 2:1 ratio) at random. The minimisation/randomisation schedule (the order in which participants are allocated by minimisation or randomisation) is prepared by a statistician within the Robertson Centre for Biostatistics (University of Glasgow) using the method of randomised permuted blocks. This statistician will not carry out the final analysis for the study. Participants will be remotely allocated by study fieldworkers using an automated telephone service operational 24 hours a day. Allocation will be performed after the participant has signed the consent form (online supplementary appendix 2) and completed relevant baseline data collection procedures. On group allocation, participants will be allocated a unique randomisation number.

## The intervention group

The intervention and logic model were developed during a 12-month development phase. Several behaviour change theories were identified during this process that underpin the intervention, including Social Cognitive Theory,[38] Control Theory,[39] Self-Determination Theory[40] and Social Support Theories.[14] This development process will be described in more detail in a complementary paper. In brief, table 1 demonstrates how the logic model's mechanisms of action (figure 1) will be operationalised within the intervention.

Overall, the website will provide evidence-based information for participants and helpers on healthy eating, physical activity and guidance on how to select and/or be a good helper (see below for details). The app will be used to set and monitor weight loss goals, and as a platform to promote interaction with nominated helpers. Example screenshots from the app and website are provided in the online supplementary appendices 3–7.

The HelpMeDoIt! intervention will be delivered via a smartphone app and website. The core aspect of the intervention involves participants nominating one or more 'helpers' from among people they know to support them with their weight loss goals. Overall, the intervention will have seven key elements, including: (i) support for goal setting and planning; (ii) 'track your progress' for monitoring; (iii) 'nominate your helper' to identify social support; (iv) obtain agreement from nominated helper(s) to provide support; (v) helper-specific advice on how to provide effective support; (vi) behaviour-specific information (including 'tips' and case stories); and (vii) the goal updates and support element of the intervention.

The app and website have different functions which complement each other; therefore, participants and their nominated helpers will be encouraged to make use of both the app and website. This combined approach was agreed with user representatives during our initial intervention development phase. The website was designed to be accessible and viewable from both desktop computers and smartphone devices. Insights from users and experts from the software company also highlighted the need for the app to focus on only a few key features. Substantial amounts of text or content on apps were identified as a barrier to engagement. Users from our development phase therefore highlighted the preference for a separate website which contained more detailed information on goal setting, self-monitoring, diet and physical activity.

The website presents evidence-based information for both participants and helpers.

► *Participant information* will include: (i) guidance on how to use the app for setting SMART goals (specific, measurable, achievable, relevant and timely), monitoring progress and identifying appropriate helpers; (ii) up-to-date information on healthy eating, physical activity and behavioural strategies to support weight loss; (iii) 'top tips' for weight loss (based on the key points of the evidence-based information, eg, 'add volume to your meal with liquid or fibre'); and (iv) 'helpful links' (which includes web links to other relevant pages).

**Table 1** Features of the app and website linked with corresponding elements of the HelpMeDoIt! logic model (figure 1)

| Logic model components | Associated app and website components | |
| --- | --- | --- |
| | **Participant** | **Helper** |
| Facilitate and encourage social support | – 'Nominate your helper' feature on app<br>– Two methods of interaction via app<br>– Guidance on website | – 'Nominate your helper' feature on app<br>– Two methods of interaction via app<br>– Guidance on website |
| Provide support to helpers | – Animated smile feature on app | – Guidance on website |
| Encourage and provide support for goal setting, action planning and problem solving | – Guidance on website<br>– Goal categories and templates on app<br>– Encouragement and advice via daily app messages/tips | – Guidance on website<br>– View participants' goals via app<br>– Encouragement and advice via daily app messages/tips |
| Facilitate and encourage self-monitoring | – Self-monitoring and progress graphs feature on app<br>– Self-monitoring guidance on website | – View participants' progress on app<br>– Self-monitoring guidance on website<br>– Helper guidance on website |
| Share tips | – Top 10 tips feature on website<br>– Encouragement and advice via daily app messages/tips | – Top 10 tips feature on website<br>– Encouragement and advice via daily app messages/tips |
| Support self-efficacy | – Self-monitoring and progress graphs feature on app<br>– Motivating messages received via app for goal progress<br>– Weekly email summary report<br>– Motivational messages from helpers<br>– Receiving animated smiles. | – Helper guidance via website<br>– Instant method of interaction via app with animated smiles |
| Boost motivation | – Self-monitoring and progress graphs<br>– In-app reward of medals/trophies for regular login and progress<br>– Encouragement via animated smiles from helper<br>– Encouragement and advice via daily app messages/tips<br>– Guidance on website | – In-app reward of medals/trophies for frequent login and input<br>– Encouragement via animated smiles from participants<br>– Helper guidance via website<br>– Encouragement and advice via daily app messages/tips |
| Provide healthy eating advice | – Guidance on website<br>– Encouragement and advice via daily app messages/tips | – Guidance on website<br>– Encouragement and advice via daily app messages/tips |
| Provide physical activity advice | – Guidance on website<br>– Encouragement and advice via daily app messages/tips | – Guidance on website<br>– Encouragement and advice via daily app messages/tips |
| Provide behavioural control/well-being advice | – Guidance on website<br>– Encouragement and advice via daily app messages/tips | – Guidance on website<br>– Encouragement and advice via daily app messages/tips |
| Facilitate encouragement, feedback and reinforcement | – Animated smiles feature on app<br>– Encouragement and advice via daily app messages/tips<br>– Messages from helpers | – Guidance on website<br>– Animated smiles feature on app<br>– Encouragement and advice via daily app messages/tips |
| Promote physical activity | – Physical activity goal category and templates<br>– Guidance on website<br>– Top 10 tips feature on website<br>– Encouragement and advice via daily app messages/tips | – Guidance on website<br>– Top 10 tips feature on website<br>– Encouragement and advice via daily app messages/tips |
| Promote healthy eating | – Healthy eating goal category and templates<br>– Guidance on website<br>– Top 10 tips feature on website<br>– Encouragement and advice via daily app messages/tips | – Guidance on website<br>– Top 10 tips feature on website<br>– Encouragement and advice via daily app messages/tips |

Continued

**Table 1** Continued

| Logic model components | Associated app and website components | |
| --- | --- | --- |
| | Participant | Helper |
| Promote overall well-being | – Well-being goal category and templates <br> – Guidance on website <br> – Top 10 tips feature on website <br> – Encouragement and advice via daily app messages/tips | – Guidance on website <br> – Top 10 tips feature on website <br> – Encouragement and advice via daily app messages/tips |
| Opportunities for social comparison and learning from peers | – Case stories feature on website (to be added after stage 2 commences) <br> – Helper interaction | – Case stories feature on website (to be added after stage 2 commences) |
| Promote autonomy | – Encourage customisation of goals <br> – Ability to add own goals <br> – Encouragement and advice via daily app messages/tips <br> – Website and app designed in an autonomy supportive way <br> – Personalisation of settings | – Guidance on website to support participant to set own goals in an autonomy supportive way <br> – Encouragement and advice via daily app messages/tips |
| Provide social support (instrumental and emotional) | – Guidance on website <br> – Encouragement and advice via daily app messages/tips | – Guidance on website <br> – Encouragement and advice via daily app messages/tips |
| Support ongoing goals around physical activity, diet and well-being | – Guidance on website <br> – Encouragement and advice via daily app messages/tips | – Guidance on website <br> – Encouragement and advice via daily app messages/tips |

► *Helper information* aims to provide helpers with the guidance needed to be an effective helper to the participant who is trying to lose weight and will include: (i) tips on how to be a good helper; (ii) methods of positive feedback and encouragement to the participant, for example, the option to send animated smiles; (iii) examples of non-food rewards to help motivate the participant (eg, new music for their friend's iPod); and (iv) examples of dialogue and motivational language to support the participant. Helpers can interact with the participants via phone call, in person, text message or by sending 'smiles' via the app which displays a range of statements such as 'Keep up the good work!' or 'Great job this week'.

The app focuses on the three key behaviour change strategies of goal setting, self-monitoring and social support.

► *The participant version of the app* will include the following features: (i) 'goal setting'; (ii) 'monitor your progress'; (iii) 'nominate a helper' (also with the option to 'remove a helper'); and (iv) methods of interacting with their helper, for example, sending animated smiles, text-based message and/or phone calls.

► *The helper version of the app* will include the following features: (i) display of the participant's goals; (ii) display of the participant's progress for weight, goals completed and smiles received; and (iii) methods of positive feedback and encouragement to the participant, for example, sending animated smiles, text-based messages and/or phone calls.

An element of 'gamification' will be used within the app to encourage frequent use and to support ongoing engagement of both participants and helpers. This will involve both participants and helpers receiving points for: (i) regular input of progress data; (ii) interaction with each other; and (iii) successful achievement of goals. Once participants and/or helpers accumulate a certain number of points they will be awarded virtual medals, that is, bronze, silver or gold or a trophy. Ongoing engagement will also be supported via the use of: (i) push and email notifications (eg, informative messages, progress summaries and notifications of new badges); (ii) daily motivating messages (eg, 'Doing well? Think about how you can progress one of your goals this week'); and (iii) weekly reminders for uncompleted goals. Helpers will be sent: (i) daily informative messages (eg, 'A kind word can do wonders for motivation. Say 'great job' to your friend today'); and (ii) be sent regular prompts via push notification and email to remind them to provide encouragement, celebration or further support.

### Nominated helpers
Participants can nominate one or more people to be an official helper. They may be family, friends or colleagues. They are not restricted to helpers from the UK. If individuals agree to be helpers, they will be directed to the study website where they will be able to access an information sheet about the study. They will be asked to signify their consent using an online form. This will indicate their consent to be a helper, for the study team to keep their contact details and also to signify whether they are

willing to be contacted to complete an interview at a later date (for which there will be a separate consent process). They will then enter brief demographic details and their contact information on the website.

The 'active' phase of the intervention will run for 6 months where participants/helpers will receive reminders to use the system. After this period, they will still be able to access the intervention until 12 months but they will no longer receive reminders from the app.

### Exploring the feasibility of participants also acting as helpers

The HelpMeDoIt! study aims to explore how the intervention might work in a real-world setting. Due to the social support focus of the intervention, it may be that two or more friends/relatives wish to lose weight together and support each other. It is important to allow for and explore this for several reasons, including: (i) this approach may have potential benefits for participants via increased support and motivation; (ii) participants who also act as helpers might have more beneficial outcomes than participants who don't act as helpers; and (iii) identifying a spillover effect in line with the diffusion of innovation theory (ie, the HelpMeDoIt! intervention gains momentum and spreads through a specific social network).[41] Our study will therefore allow participants in the intervention arm to act as a helper for a friend/relative. Their helper will then also have access to the participant aspect of the intervention (ie, so that they can be both participants and helpers to each other). However, to avoid contamination of the findings the second individual will not be registered as a 'study participant' or randomised. If they were to be registered as a study participant, this could potentially contaminate the randomisation (ie, one individual may be randomised to the intervention group and the other to the control group). It is unknown if participants will choose to act as helpers but it is important to allow for and explore this as part of the feasibility study.

### The control group

The control group will receive a leaflet about the health benefits associated with healthy eating and physical activity behaviour change. They will not receive any social support or personalised content. Participants in the control group are not restricted in any way regarding their involvement in other weight loss activities. They can continue to embark on weight loss strategies, for example, join a slimming club. Controls can access the website and app after follow-up is complete at 12 months.

### PROGRESSION CRITERIA FROM FEASIBILITY TO FULL TRIAL

The feasibility of the trial methods, the feasibility and acceptability of the intervention, and its potential to be further developed and delivered in a full randomised controlled trial are the key outcomes of this study. Feasibility will be assessed using the progression criteria outlined in table 2. These criteria have been finalised

within our Trial Management Group and approved by our Trial Steering Committee. Final assessment of the progression criteria will be undertaken by the Trial Steering Committee following analysis of the findings. Multiple methods are employed to assess feasibility, including: (i) outcome measures; (ii) process evaluation measures; and (iii) an economic evaluation. There was substantial debate around criterion 6. On the one hand, current evidence on app usage indicates that around 25% of users will engage with an app only once.[42] However, if only a minority of participants engage with the app and it is effective for them, then it may have a cost-effective and worthwhile impact on public health. On the other hand, we would want to see a reasonable proportion of participants engaging with the app sufficiently to set goals and identify helpers, even if the subsequent interactions with their helpers are not made via the app.

### Outcome measures

A full list of measurable outcomes is presented in table 3. Measures will be completed face to face with a study researcher, with the exception of one telephone-based outcome measure, in the participant's home or an interview room in the university. All staff involved in data collection will be given training in study procedures, attend Good Clinical Practice training and hold a National Health Service (NHS) Research Passport from NHS Greater Glasgow and Clyde.

### Primary outcomes

Three primary outcomes will be assessed: BMI, physical activity and diet (table 3). Each will be measured at baseline and 12 months. We will assess which of these is most feasible for a future full trial. Since measuring diet[43] and physical activity[44] in community-based trials is challenging, we will assess two ways of measuring these outcomes. This will help inform the choice of primary outcome for a future full trial.

BMI ($kg/m^2$) will be calculated from measures of height and weight. Height will be measured using a Seca Leicester Height Measuring Stadiometer, with participant facing forward, wearing no shoes and with their head in the Frankfort Plane (parallel to the floor). Measurements will be recorded once, in cm, to one decimal point. Weight will be measured, in the absence of shoes, using Tanita HD 352 High-Capacity Low-Profile Electronic Weighing Scales. Scales will be calibrated before first use. Weight will be recorded once, in kg, to one decimal point.

Physical activity will be measured using Actigraph GT3X accelerometers, objective activity monitors which measure duration, intensity and frequency of physical activity. Participants will be asked to wear the accelerometer on their right hip for 7 days during waking hours (except when swimming or bathing). Participants will receive their accelerometer during a face-to-face visit allowing correct placement of the device. This will be demonstrated by the researcher. Data will be collected in 1s epochs, at a sample rate of 100 Hz, and converted

| Table 2 | Progression criteria from feasibility trial to full randomised controlled trial |
| --- | --- |
| **Progression criterion** | **Method of assessment** |
| 1. Is the intervention feasible to deliver and acceptable to participants and their helpers? | ▶ USE questionnaire<br>▶ Participant/helper interviews |
| 2. Are participants willing to be randomised to the intervention? | ▶ Recruitment experiences of the study team and fieldworkers<br>▶ Insight from qualitative interviews with participants |
| 3. Are appropriate and effective routes of recruitment available to achieve a powered sample size in a full trial? | ▶ Coming close to the sample size, as judged by the Trial Steering Committee (TSC), with reasonable expectations of being able to address any recruitment issues |
| 4. Are identified barriers and challenges to implementation of the intervention planned for and surmountable? | ▶ Process evaluation which will present a SWOT analysis (Strengths, Weaknesses, Opportunities and Threats) and action plan |
| 5. Are appropriate retention rates achieved at 12-month follow-up? | ▶ Measured using the following scale in both the intervention and control group at 12 months: if >70% followed up, proceed; if 50%–69% followed up, discuss with TSC; if <49% followed up, do not proceed |
| 6. Do the majority (>50%) of participants within the intervention group visit the app at least twice or do 25% of participants randomised use it three or more times? | ▶ App usage statistics and/or participant interviews |
| 7. Do the data collection procedures effectively collect the data required for a full trial? Successful completion of *at least one* data collection method (BMI, physical activity *or* healthy eating) at both baseline and at 12 months in those retained measured using the following scale: | ▶ If >90% *of at least one* data collection measure completed, proceed<br>▶ If 70%–89% *of at least one* data collection measure completed, discuss strategies for improvement in future trial with TSC<br>▶ If <70% *of all three* data collection measures completed, do not proceed without further modification and pilot |
| 8. Are the intervention costs of a full trial covered? | ▶ Identification of a source to pay access and treatment costs |

to 15 s epochs for analysis using Actilife-6 software.[45] Non-wear time will be identified by >60 min of continuous 0 counts and removed before analysis. Data will be included for analysis where the accelerometer has been worn for a minimum of 4 days, and with a minimum wear time of 10 hours per day. Freedson cut-points[46] will be used to determine the amount of time spent sedentary and in moderate-to-vigorous physical activity.

The 7-day Physical Activity Recall questionnaire[47] will be used to subjectively measure physical activity. Participants, guided by the researcher, will self-report their activity over the previous 7 days in relation to moderate, hard and very hard exercise. This measure has been validated for use in adult populations,[48] and researchers will adhere to the protocol published by Sallis *et al*.[47] Two methods of physical activity measurement are being explored to assess their feasibility and usability for a future trial.

Diet will be measured by the Dietary Instrument for Nutrition Education (DINE) questionnaire,[49] a validated 7-item questionnaire to explore the frequency of consumption of different food types, for example, bread and rolls, cereals and meats. Fieldworkers will ask participants to report the frequency with which they eat specific foods. The frequencies will be scored using DINE guidelines[49] to produce an overall score for fat and fibre. Diet will also be measured via repeat 24 hours dietary recall,[50] collected by a researcher via telephone on four separate days within a

10-day period (including one weekend day). Participants self-report their food intake, prompted by the researcher, for the previous 24 hours. Researchers will be guided by photographic and textual examples of portion sizes, which they can use as prompts over the phone. Participants food intake will be inputted onto dietary analysis software[51] and analysed for energy intake, macronutrients and fibre. Two methods of assessing diet are being explored in order to assess their feasibility for a future trial.

### Secondary outcomes

Waist circumference will be measured using a 2 m flexible tape measure with buckle, around the midpoint between the iliac crest and inferior margin of the lower rib. Hip circumference will be measured around the widest point of the buttocks. Measurements will be recorded twice in cm to one decimal point (eg, 95.2 cm). A third measure will be taken if the difference is >0.5 cm.

Health-related quality of life will be measured using the EuroQol five-dimensions (EQ-5D) questionnaire and quality-of-life thermometer.[52] This measure is used frequently in health-related research to explore five dimensions: mobility, self-care, usual activities, pain and anxiety/depression. An additional measure of capability well-being will be measured using the ICEpop CAPability measure for Adults (ICECAP-A)[53] scale. This is a new

**Table 3** Outcome measures

| Demographics | | |
|---|---|---|
| Case report form: gender, age, socioeconomic status, employment and education status, current weight loss status, current health status, current computer and phone use | | Baseline and 12 months |
| *Primary outcomes* | | |
| Body mass index (kg/m$^2$) | Physical measurement of height (m) and weight (kg) | Baseline and 12 months |
| Diet | DINE questionnaire[39]<br>4 days of 24 hours dietary recall[40] | Baseline and 12 months |
| Physical activity | 7-Day accelerometry[36]<br>7-Day Physical Activity Recall Questionnaire[37] | Baseline and 12 months |
| *Secondary outcomes* | | |
| Anthropometric changes | Waist and hip circumference (cm) | Baseline and 12 months |
| Health-related quality of life | EQ-5D questionnaire[41]<br>ICECAP-A scale[42] | Baseline and 12 months |
| Mental health | General Health Questionnaire[43] | Baseline and 12 months |
| National Health Service resource use and participant-borne costs | Specially designed resource use questionnaire | Baseline and 12 months |
| Usability of software | USE questionnaire[29] | 12 months |
| Smoking use | Heaviness of Smoking Index[44] | 12 months |
| Alcohol use | Alcohol Use Disorders Identification Test[45] | 12 months |
| *Mediators of change* | | |
| Social support | Exercise & Eating Habits Social Support Scales[46] | Baseline and 12 months |
| Self-efficacy | Weight[47] & Exercise Efficacy Lifestyle Scales[48 49] | Baseline and 12 months |
| Motivation | Treatment Self-Regulation Questionnaire[50] | Baseline and 12 months |
| Social networks | Sociogram[51]<br>Egocentric questionnaire[51] | Baseline and 12 months |

scale which, compared with the EQ-5D, explores less clinically related changes in quality of life over four dimensions: feeling settled and secure; being independent; achievement and progress; and enjoyment and pleasure. This might be an appropriate measure in our population due to the potentially large range of participant characteristics. Mental health will be measured using the General Health Questionnaire,[54] a validated and frequently used 12-item self-report questionnaire.

We will gather data on NHS resource use and participant-borne costs using a specially designed resource questionnaire. These data will help us establish key cost drivers of the intervention.

At 12 months, we will use the Usability, Satisfaction and Ease of Use (USE) questionnaire[55] to assess feasibility and acceptability of the app and website. We will also use the Heaviness of Smoking Index[56] and Alcohol Use Disorders Identification Test[57] questionnaires at 12 months using this opportunity to assess the feasibility of additional questionnaires for data collection. These may be helpful in identifying other potential lifestyle changes made my participants in a future trial related to 'spillover' effects of the intervention.

## Process evaluation measures

The process evaluation will explore in detail the way in which the intervention operates to produce outcomes, as well as to assess feasibility and acceptability. The evaluation will be conducted based on MRC guidelines for process evaluations of complex interventions[58] and will examine the following elements: (i) context; (ii) fidelity of the intervention; (iii) exposure to the intervention; (iv) reach; (v) recruitment and retention; (vi) contamination; (vii) the control arm; and (viii) mechanisms of impact. Table 4 presents the multiple points of data collection for the process data. In brief, some quantitative data will inform the process evaluation (eg, intervention usage statistics). The remaining process data will be gathered via qualitative interviews with participants and helpers, and questionnaires exploring mediators of change (details below).

### Qualitative interviews with participants

At 6 months, we will interview up to 30 participants (depending on data saturation). Participants will be purposively sampled for a range of characteristics (eg, level of app/website use, age, gender). We will also specifically seek to interview those who did not take up the intervention to explore the reasons for this. At 12 months, we will also interview up to 20 participants (10 participants who continued to use the website/app after 6 months, and 10 who ceased to use it). Semistructured interview guides will

**Table 4**  Process evaluation measures

| Measure | Example questions to be answered | Method |
|---|---|---|
| Context | What type of phone/platform did participants use, for example, Android, iOS?<br>How did participants/helpers access the website for example, on phone, on desktop, on tablet?<br>Were there any differences between social networks and what impact did these have on outcomes? | Quantitative analysis<br>Social Network Analysis<br>Qualitative interviews with participants and helpers |
| Fidelity | Was the intervention delivered as intended?<br>When, if any, were any adaptations needed to the planned intervention? | Descriptive analysis of data usage statistics from software company<br>Qualitative interviews with participants and helpers, and study team |
| Exposure | How often did participants/helpers use the app and website?<br>How often did participants/helpers access the goal setting/monitoring feature of the app and website?<br>How often did participants/helpers interact with each other via the app and website?<br>What were the patterns and trends of usage over time? | Analysis of data usage statistics from software company |
| Reach | How well does the study sample represent the population of interest?<br>To what extent did the intervention reach and influence people other than recruited participants, including helpers?<br>What were the particular difficulties/issues that arose during the study in delivering the intervention? | Descriptive statistics<br>Qualitative interviews with participants and helpers |
| Recruitment and retention | What are the difficulties in recruitment?<br>What is the attrition rate overall and by group?<br>What venues do participants chose to meet for their fieldworker appointments?<br>What are the reasons for withdrawal?<br>What factors were involved in ongoing engagement with the intervention? | Quantitative analysis<br>Qualitative interviews with participants and helpers |
| Contamination | What are the characteristics of other groups' people are attending, for example, slimmer's groups?<br>Have any of the control group seen intervention content from other participants or acted as a nominated helper? | Quantitative analysis<br>Qualitative interviews with participants and helpers |
| Control arm | What is happening in the control arm? | Qualitative interviews with participants |
| Mechanisms of impact | What role does social network play in how participants use the intervention?<br>How did participants perceive their social support for healthy eating and physical activity changed throughout the intervention? | Mediators of change questionnaires (see table 3) |

be used at both time points to explore participant insights related to acceptability of the outcome measures, acceptability and usability of the app and website, patterns of usage, impact of the intervention on behaviour, support received from helpers and barriers to use. Mediators of change will also be explored in the qualitative interviews.

### Qualitative interviews with helpers
At 6 months, we will interview up to 20 helpers, purposively sampled for a range of characteristics (eg, level of app/website use, age, gender). A semistructured interview guide will be used to explore helper insights related to acceptability, guidance provided for being a helper, types of support provided to their friend, challenges of supporting their friend and/or using the app and website,

and changes in their own health behaviour as a result of being a helper.

Interviews will be completed by trained researchers via the telephone or face to face at a preferred venue (eg, their home or university). Interviews will be audio recorded and transcribed verbatim. A separate informed consent process will take place for the qualitative interviews.

### Questionnaires exploring mediators of change
Exploratory analysis of mediators is important to identify the processes by which the intervention brings about change. These will help to further refine the logic model for a future full trial. Data on social support, self-efficacy and motivation will be collected at the same time as the main outcome data using the following questionnaires

respectively: Exercise & Eating Habits Social Support Scale,[59] Weight[60] & Exercise Efficacy Lifestyle Scales[61 62] and the Treatment Self-Regulation Questionnaire.[63]

### Social network data

We will explore the characteristics of participants' social networks at the beginning of the intervention and at the 12-month follow-up using a Social Network Analysis.[64] Participants will be asked to draw a sociogram of their own 'ego' social network, highlighting various elements such as positive and negative influences and network density. Participants will also complete a questionnaire to explore additional characteristics of the individuals they intend to nominate as their helper(s), for example, frequency of contact and type of relationship. All data on participants' social networks will be gathered anonymously via the use of initials (no names will be collected).

### Web/app analytics

We will collect app and website usage data for both helpers and participants using Google Analytics to assess engagement with the intervention. Key usage data includes number of logins to the website and app by helper and participant, duration of login, average sessions per user, pages viewed and how often, whether participants set goals and entered weights, number of helpers nominated, contacts between helpers and participants via the app, number of views of 'progress charts' by participant and helper, and patterns of use over time.

### Economic evaluation

The economic analysis aims to identify and measure the key cost drivers of the intervention and control arms as well as identify suitable outcome measures for a future economic evaluation. A costing exercise will be undertaken to provide an indication of the direct costs of the intervention. This will involve monitoring all resources used in delivering the intervention and valuing them in relevant units. In addition to this, an estimation of any intervention effects on NHS and Personal Social Services (PSS) resource use (eg, GP visits) and personal costs (eg, gym membership and food purchases) will be collected via a specially designed resource use questionnaire. Together, these will indicate the relative importance of the economic evaluation in any future trial. A value of information analysis[65] will further provide information on the likely return on investment in the intervention. The economic analysis will assess the feasibility of using the EuroQol EQ-5D instrument[52] and the ICECAP-A[53] instrument as a means of capturing any short-term effects the intervention may have on health-related quality of life and/or capability well-being. The economic evaluation will adhere to guidelines for good economic evaluation practice as outlined by Gold *et al*.[66] Specific guidance will also be sought from the National Institute for Health and Care Excellence economic evaluation public health reference case[67] since it is anticipated that this intervention is likely to impact costs and outcomes beyond the NHS and PSS and thus require a broader public sector evaluative perspective.

## QUANTITATIVE ANALYSIS

A detailed Statistical Analysis Plan has been drafted and will be finalised before the study statisticians are given access to the study group allocations.

► *Baseline characteristics* will be summarised overall and by randomised group. Participant characteristics will be summarised in relation to socio-demographic, lifestyle, occupational, health status and quality-of-life variables.

► *Feasibility measures* will be the primary focus of the analysis. Follow-up rates at 12 months will be reported overall and by randomised group, with 95% CIs. The association between baseline factors and follow-up will be assessed using logistic regression, with follow-up (yes/no) as the response variable. A multivariable regression model will be developed to identify independent predictors of follow-up. Use of the intervention will be summarised for the intervention group, overall and in relation to selected baseline characteristics. The availability and utility of data relating to data usage for the app and website will be explored, and a range of summary measures will be presented in the final statistical outputs.

► *Efficacy outcomes* will be summarised overall and by randomised group, and compared using linear regression models, with randomised group, the baseline measurement of the outcome, age and gender as predictor variables. Regression models will also adjust for the minimisation factors. The residuals from each regression model will be assessed for normality. Where necessary, the outcome measure (at follow-up and at baseline) will be transformed to improve model fit. All analyses will be conducted under intention-to-treat principles and complete case analysis used, unless >20% of cases are lost due to missing data, in which case multiple imputation will be performed. These analyses are exploratory and underpowered, so no formal hypothesis testing will be performed but effect sizes will be reported in line with Consolidated Standards of Reporting Trials guidelines for reporting feasibility and pilot studies (http://www.bmj.com/content/355/bmj.i5239).

► *Process evaluation measures* will be descriptively analysed to summarise use of the app and website. A per-protocol analysis will also be conducted using simple proxies for adherence (eg, website login/% of webpages accessed) in order to identify the treatment effect associated with adherence.

► *Potential mediators of change*, such as social support, motivation and self-efficacy, will be used in an exploratory mediation analysis to assess whether they might lie on the causal pathway and to test the logic model.[68] This will assist in the decision as to whether we need the mediation measures in a future full trial.

- *Social Network Analysis* will be performed on participant sociograms and egocentric questionnaires. Sociogram data will be captured using pen and paper and later transferred to an ego-network software package such as VennMaker or EgoNet. Data will then be exported for analysis. We will calculate network measures on the alter level (eg, homophily) and the network level (size, density, EI-Index, diversity, components, proportions of ties with specific attributes). Multivariate analyses will then be run on both alter and network level data.

- *The cost data* will be summarised and described using mean values and variation around these estimates. Key fixed and variable costs of developing the intervention will be described and summarised. EQ-5D and ICECAP-A outcome data will be reported by within-attribute response rates, mean values and associated variance. Within-trial economic analyses will be performed using STATA V.12.0 and reported in line with recent CHEERS guidelines[69] and the UK public health reference case.[70] Missing data will be handled using multiple imputation (for both cost and outcome data).[71 72] Cost–utility estimates will be presented on a cost-effectiveness plane using the UK's threshold for willingness to pay for an incremental quality-adjusted life-year.

## QUALITATIVE AND MIXED-METHODS ANALYSIS

- *Qualitative data analysis* will explore the acceptability of the intervention, the extent to which participants and helpers engaged with it, perceptions of how the intervention influenced behaviour, the value of helpers' support and contextual factors. Qualitative data will be analysed by two researchers who will independently code using Braun and Clarks' approach to thematic analysis.[73] The resulting coding framework will be discussed between researchers and also within the larger study team to finalise meaningful themes and subthemes. Twenty per cent of the interviews will be double coded. Disagreements will be resolved by discussion. The analyses will test the hypothesised causal pathways expressed in the logic model and will also develop the intervention's theory of change where little is currently known (eg, how social support and web-based goal monitoring operate together to change health behaviours). This will inform the study design of any future trial by determining which elements of the intervention work well for health behaviour change in participants, how they interact with each other and which need adjustment or further development.

- *Analysis of the logic model* will be conducted using the following mixed-methods data integration strategies. Qualitative data, intervention usage statistics and Social Network Analysis data will be collected and analysed as separate data sets in the first instance. The three sets of data will then be mapped onto the logic model and triangulated with each other for complementarity (where data build up a more integrated picture of how the intervention works) and also to identify any disagreements between data sets. Depending on time and resources, the following steps will be undertaken to resolve potential disagreements in data: (i) disagreements between sets of data will be resolved using the four strategies identified by Pluye *et al*,[74] and (ii) any questions arising from a data set at this point an assessment will be made about whether these question(s) could be resolved or themes further explored by interrogating data from another data set (the 'following a thread' method).[75] An 'interpretive rigour' checklist[75] will be used to optimise the rigour of interpretation and meta-inferences made in producing the final logic model drawing on the three data sets.

All quantitative analyses will be performed in SAS for Windows V.9.3 and/or R for Windows V.3.2.2, or higher versions of these programs. Qualitative analyses will be performed using NVivo 10.

### Sample size
We intend to recruit 120 participants. Since we are most interested in the intervention, we will recruit using a 2:1 ratio with 80 participants in the intervention group and 40 in the control group. We expect a dropout rate of 30%. This final sample size of 84 for analysis is not powered to detect any differences between groups for the proposed effectiveness outcomes (BMI, physical activity and diet) but will provide enough precision to estimate any feasibility proportion (eg, proportions retained/found the study acceptable/provided outcome data) across the whole sample to within ±11 percentage points using a 95% CI. This would also allow for the estimation of the mean of a continuous outcome (such as BMI) in the intervention arm to within 0.262 of an SD.

### Data management
To ensure safe and accurate data management, the study team will adhere to the agreed HelpMeDoIt! Data Management Plan. All study data will be gathered by trained researchers using hardcopy paper questionnaires and stored in a locked filing cabinet within our secure research unit. Data will then be entered by approved study personnel onto a secure online database (developed and hosted by the Robertson Centre for Biostatistics, University of Glasgow). Data will be entered using a unique participant ID so that study personnel remain blinded to group allocation. Appropriate elements of the database have automated error checking facilities to ensure only valid data are entered. We will perform single data entry, of which 10% will be cross-checked by double entry. Full double data entry will be performed if >5% error rate is detected. The study team and staff performing data entry work closely together within the same research unit, enabling quick identification and rectification of any errors. Qualitative data will be anonymised in all interview transcripts.

Several strategies will be employed to minimise sources of data bias, including random allocation of participants, retention methods to reduce loss to follow-up, restriction of only one participant per household and researchers completing data collection not involved in either delivery of the intervention or data analysis. All data will be kept for 10 years in line with University of Glasgow Research Governance Framework Regulations for clinical research. These data will be stored confidentially on password-protected servers. The final data set will be accessible by approved members of staff from the research team and Robertson Centre for Biostatistics, University of Glasgow. Approved members of the software company, who have signed a Data Protection Agreement, will have access to limited participant contact details to enable them to effectively manage software errors. The low-risk nature of this study means that a Data Monitoring and Ethics Committee (DMEC) is not required for this feasibility trial. Our Trial Steering Committee will cover the functions of the DMEC, particularly in relation to ethical issues, patient safety and continuation of the trial.

### Data sharing

We will implement the following data sharing policy. Participants will have the option of consenting to the research team sharing their data with other researchers. This would involve their data being stored anonymously with the UK Data Archive (an internationally acknowledged centre of expertise who store research data for use by researchers and scientists). Other genuine researchers may then access these data to help answer future research questions. All information stored adheres to the Data Protection Act 1998. Participants will never be identifiable from the research data. Participants *do not* have to give consent to data sharing to be able to take part in the study.

### Retention of participants

We will attempt to maintain good participant retention using several methods, including giving options for participants regarding where data collection takes place, provision of newsletter updates, sending birthday cards, obtaining mobile numbers and alternate contact details, offering a reduced item version of the follow-up questionnaire to provide at least a minimum data set for participants who are reluctant to complete the full follow-up and £20 voucher payments as a thank you for each point of data collection.

Participants have the right to withdraw consent for participation in the HelpMeDoIt! study at any time. If a participant initially consents but subsequently withdraws from the study, clear distinction will be made as to what aspect of the study the participant is withdrawing from. This could be: (i) withdrawal from the intervention; (ii) withdrawal from follow-up data collection; or (iii) withdrawal from entire study. Any retrospective request for data to be deleted will be respected.

### ETHICS AND DISSEMINATION

Ethical approval for stage 1 was granted by the University of Glasgow MVLS College Ethics Committee (Ref: 200140108). Ethical approval for stage 2 has been granted by the NHS West of Scotland Research Ethics Committee (Ref: 15/WS/0288). Research governance approval has been given by NHS Greater Glasgow and Clyde Health Board. The study will be conducted in accordance with the recommendations for physicians involved in research on human participants adopted by the 18th World Medical Assembly, Helsinki 1964 and later revisions. Findings from this study will be disseminated through multiple peer-reviewed publications and conference presentations. We also intend to participate in public engagement events, for example, via the Glasgow Science Centre.

### ASSESSMENT OF HARMS

The intervention is low risk to participants. There is a risk that participants may set unhealthy goals and that helpers may not provide support in a positive way. However, we will provide guidance to helpers to ensure that they are aware of how to provide positive support for participants. We will include information about healthy diet in line with government recommendations as well as advice on safely increasing physical activity levels. Participants will be encouraged to discuss any health concerns with their GP who will be informed of their participation. We will encourage fieldworkers, participants and helpers to report negative outcomes or experiences to the study team (email, telephone or by post) and we will explore the issue of 'harm' in the interviews with both participants and helpers.

**Author affiliations**
[1]MRC/CSO Social and Public Health Sciences Unit, Institute of Health and Wellbeing, University of Glasgow, Glasgow, UK
[2]College of Engineering, Mathematics, and Physical Sciences, Data Science Institute, University of Exeter, Exeter, UK
[3]Robertson Centre for Biostatistics, Institute of Health and Wellbeing, University of Glasgow, Glasgow, UK
[4]Health Economics and Health Technology Assessment Unit (HEHTA), Institute of Health and Wellbeing, University of Glasgow, Glasgow, UK
[5]Psychology Applied to Health (PAtH) Group, Institute of Health Research, University of Exeter Medical School, Exeter, UK
[6]Centre for the Development and Evaluation of Complex Interventions for Public Health Improvement (DECIPHer), Cardiff University, Cardiff School of Social Sciences, Cardiff, UK
[7]Division of Population Medicine, School of Medicine, Cardiff University, Cardiff, UK
[8]Centre for Trials Research (CTR), School of Medicine, Cardiff University, Cardiff, UK

**Acknowledgements** The HelpMeDoIt! team would like to thank their software collaborators at JamHot for their development and hosting of the app and website; participants in the stage 1 development phase who helped finalise the HelpMeDoIt! intervention; and colleagues Dr Mark McCann, Dr Robert Young and Dr Heide Weishaar, for their valuable contribution to the Social Network Analysis methods.

**Contributors** LM is trial manager and responsible for coordinating the HelpMeDoIt! study. LM was involved in finalising the study protocol, implementing study processes and drafting the manuscript. JP was involved in finalising the study protocol, in particular the Social Network Analysis methods and qualitative analyses, and reviewing the manuscript. OU is trial administrator and was involved in finalising the study protocol, in particular in relation to intervention content, and reviewing the manuscript. LMo, EC, KH, SM, was involved in finalising the

study protocol and reviewing the manuscript. MK was involved in finalising the study protocol, in particular the statistical considerations, and reviewing the manuscript. SMT were involved in finalising the study protocol, in particular the process evaluation methods, and reviewing the manuscript. AM was involved in finalising the study protocol, in particular the statistical analyses, and reviewing the manuscript. EM was involved in finalising the study protocol, in particular the health economics analysis, and reviewing the manuscript. SS is principal investigator and responsible for overall management of the HelpMeDoIt! study. She led the design of the intervention, the development and finalisation of the protocol and reviewed the manuscript.

**Funding** This work was supported by the National Institute of Health Research (Public Health Research Programme grant number 12/180/20). Additional time on the study was supported by UK Medical Research Council and Chief Scientist Office core funding as part of the MRC/CSO Social and Public Health Sciences Unit 'Social Relationships and Health Improvement' programme (MC_UU_12017/11 and SPHSU11) and 'Complexity in Health Improvement' programme (MC_UU_12017/14 and SPHSU14). SS was supported by MRC Strategic Award MC_PC_13027.

**Competing interests** None declared.

**Ethics approval** West of Scotland NHS Research Ethics Committee (Ref: 15/WS/0288) and the University of Glasgow MVLS College Ethics Committee (Ref: 200140108)

**Provenance and peer review** Not commissioned; externally peer reviewed.

**Data sharing statement** None.

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
