## [Reviewer comments · BMJ Open]

ARTICLE DETAILS

TITLE (PROVISIONAL)	Study protocol for the 'HelpMeDolt!' randomised controlled feasibility trial: an app, web and social support based weight loss intervention for adults with obesity.
AUTHORS	Matthews, Lynsay; Pugmire, Juliana; Moore, L; Kelson, Mark; McConnachie, Alex; McIntosh, Emma; Morgan-Trimmer, Sarah; Murphy, Simon; Hughes, Kathryn; Coulman, Elinor; MacAskill, Olga; Simpson, Sharon

VERSION 1 - REVIEW

REVIEWER	Mitch J Duncan University of Newcastle, Australia
REVIEW RETURNED	24-May-2017

GENERAL COMMENTS	The study described is innovative in its application of social network features to support behaviour change and is a welcome addition to the literature. The description of the trial intervention is however an area that requires revision. There is too little detail provided to clearly articulate what the intervention is and how it will achieve its goals. It is a weight loss trial although no details are provided on the strategies, beyond goal setting and social support, that are provided to participants to help achieve weight loss. For instance it would be expected that information on how to create a energy deficit to promote weight loss would be detailed, and also how participants will go about achieving this. Further, the details on what behaviours and how they are self monitored is lacking and requires addition. The authors state that another manuscript will describe the intervention but without this information in the current manuscript it is difficult to assess the characteristics of the intervention and how it will achieve its stated aims. The authors are encouraged to also include screen shots of the intervention Further comments are below. Table 1 criteria 6. What is the rationale for this criteria around usage? Usage criteria appear to be low criteria for usage of the intervention over the intervention period. Whilst the usage-behaviour change relationship may not be linear it would be expected that a higher criteria is used for this, as currently it appears that if 25% of the intervention group logged some aspect of behaviour in the app in the first few weeks it is a 'success' however it also means that a substantial proportion of the participants may not use the app – receive the intervention – for the majority of the intervention period. Outcome measures the active phase of the intervention is 6 months, yet assessments are not conducted until 12 months. Would it be useful to assess outcomes, even a subset if there are concerns of participant burden, at 6 months also?
---

	Strengths and limitations – mixed methods, should measured weight be listed also? P5. Line 35 -36. Specify what exactly is being referred to when these technologies are referred to as effective? Is this activity, diet, weight? P5 when discussing the rationale for the use of social support it would appear relevant to cite Maher et al https://www.ncbi.nlm.nih.gov/pubmed/24550083 as many of the same issues are discussed. methods. P8. No upper limit is placed on BMI, what is the likely impact of recruiting severely obese individuals who may have different behaviour change and weight change trajectories? Methods recruitment of males – why only target males through one of the recruitment strategies? Given the acknowledged difficulties in recruiting men would it not be advisable to target men via all strategies. Provide details of the recruitment pack posted to participants via the SPCRN, specifically details on how the study is introduced to these potential participants and how they were identified. Will details of the number of participants who opt for either in home assessments or the research lab assessments be reported on? This is an interesting feasibility issue. Are incentives provided to participants for completing assessments. The website and app have different functions (p11 L16), the inclusion criteria are only for a smartphone and internet access. It is possible that for some individuals internet access is by the smartphone only. Therefore is the website constructed using a responsive design approach which will be viewable on such a device to optimise viewing and usage? P11 L32. Given the app is used to set goals and self-monitor would it not be useful for people to have access to this information on the app? Is the helper positive reinforcement limited to sending 'smiles'? can they provide text based support in the form of comments? Although it is an emerging area, one might imagine that a participant may value a message 'great work keep it up' more than a animated smile? It is unclear if this is what is referred to as the motivational language on p12L5. Further how is this motivational language operationalised in the app? Description of the helper intervention content related to how it is implemented in the intervention is necessary. Currently unclear how many aspects are implemented. P12. App content. More detail is necessary on how these aspects are operationalised and implemented in the intervention. what other methods of interacting are available. Is there any opportunity for two way 'live' communication such as messaging? More detail is necessary on the methods used to prompt engagement? Does a participant get prompted to enter data after a single day of non entry? Detailing this has important implications for understanding any non-usage attrition that occurs in the trial. Is there any geographic restriction to where the helpers can be located? Inclusion criteria is trying to lose weight and there are no exclusion criteria regarding recent weight loss. How will the trial deal with people who have already lost a significant amount of weight (eg 4.5kg) or are currently using another weight lose strategy. Measures Protocol for accelerometer data. Please justify the choice of 15 second epoch, particularly as the unit used can collect raw accelerometer data which is now encouraged as it allows greater
--	---

	flexibility in the analysis of the data. Further, details should be presented on the methods used to process the data rather than say they will be developed in the future. The following articles maybe useful in guiding some of these decisions and also providing insight into how the details should be reported Montoye et al., doi:10.1136/bjsports-2015-095947 What access to the app will the helpers have? Will this be complete access and able to view a participants whole self-monitoring data? Also provide clarification around the sample size justification. The main outcomes are BMI, physical activity and diet which are presumably continuous measures so do the authors mean they are powered to detect an 11 point percentage difference in these scores ? Also as both accelerometer data and self report are collected for PA how is the decision made regarding if the intervention successfully change PA if both these measures are not significant? Consider expanding the analysis of usage data from basic descriptive to examine patterns and trends over time (eg survival analysis) as this may provider greater insight into how usage may influence behaviour change
--	---

REVIEWER	Marie LÖf Karolinska Institutet, Sweden
REVIEW RETURNED	26-May-2017

GENERAL COMMENTS	 1. The introduction should be revised in order to more clearly focus on existing literature regarding mHealth and weight loss trials and how this trial fits there. 2. Please revise so that the aim in the abstract and the aim in the end of the introduction are more consistent. The aim in the introduction does not mention weight loss. 3. Why targeting the wide age range of 18-70 years? A 18-year-old may have very different needs than a 70-year-old in regards to a weight loss app intervention. 4. Why keep women that become pregnant in the study? Weight data cannot be used and also other variables may be difficult to assess. 5. What is the rationale for doing a combined intervention with a website and an app? Why not do only an app since they today can contain a lot of information and it would make it easier with only one medium for the participants? 6. The authors should describe and motivate the dietary outcomes that will be assessed in detail. 7. Why not standardise weight measurements to include no clothing and preferably before breakfast? 7. The authors should provide some more details on the Actigraph measurements e.g. sampling frequency, and filter to be used for analysis. And why choose the waist? Have you considered wrist-worn? 8. Why two methods for dietary intake? Have the methods been validated in obese subjects? How will portion sizes be assessed? 9. Should the study protocol adhere to the CONSORT-EHEALTH: improving and standardizing evaluation reports of Web-based and mobile health interventions. J Med Internet Res 2011;13(4):e126? 10. Minor: the authors' contribution paragraph is very long.
--

VERSION 1 – AUTHOR RESPONSE

REVIEWER 1

Feedback 1: No details are provided on the strategies, beyond goal setting and social support, that are provided to participants to help achieve weight loss.

Author response 1: Thank you for this feedback. Relevant to this point and many of the specific points below, we have revised the methods section (p7 onwards) to increase the level of detail provided on the intervention, while remaining mindful of the word count allowed. To help with this, we have included a table summarising how each element of the logic model is operationalised within the intervention.

Feedback 2: The details on what behaviours and how they are self monitored is lacking and required addition.

Author response 2: As above.

Feedback 3: The authors state that another manuscript will describe the intervention but without this information in the current manuscript it is difficult to assess the characteristics of the intervention and how it will achieve its stated aims.

Author response 3: We acknowledge your important point and we have now included as much additional detail on the intervention as possible within the word limit. A complementary paper currently in preparation will provide further justification of the intervention and the related iterative development process. We believe this additional manuscript is an important paper which will be useful to other researchers developing similar interventions; however, it will include retrospective data, and a level of detail on the intervention's development that would not be appropriate or useful in a protocol paper.

Feedback 4: Include screenshots of the intervention.

Author response 4: Thank you for this suggestion. This is a good idea and screenshots have now been added as supplementary information to the paper.

Feedback 5: Table 1 Criteria 6 – what is the rationale for this criteria around usage?

Author response 5: We have justified this in the text (see page 19).

Feedback 6: Would it be useful to assess outcomes, even a subset, if there are concern of participant burden, at 6months also?

Author response 6: This issue was discussed in detail with our funder, Trial Management Group and Trial Steering Committee. Our final decision, to collect follow-up data at 12-months only, was based on the following: (i) as a feasibility study we are interested in assessing the feasibility of retaining participants for a long term follow-up; (ii) we will gather qualitative feedback from participants at 6-months providing us with valuable insight on the study; (iii) short term behaviour change for weight loss at 6-months has been demonstrated by other studies and typically doesn't relate to longer term sustained behaviour change; and (iv) we therefore chose to not add additional participant burden at the 6-month mark.

Feedback 7: Strengths and limitations- should measured weight be listed also (see p4).

Author response 7: The sentence has been amended to now include physical measurements.

Feedback 8: P5 line 35-36 – specify what exactly is being referred to when these technologies are referred to as effective?

Author response 8: This has now been clarified in the text (p6). Thank you.

Feedback 9: Social support – it would appear relevant to cite Maher et al

Author response 9: Thank you for suggesting this useful reference. It did cover relevant material and

has now been cited in the Introduction (see p5).

Feedback 10: P8 – no upper limit placed on BMI, what is the likely impact of recruiting severely obese individuals who may have different behaviour change and weight change trajectories?

Author response 10 : Due to the nature of apps being accessible to the full population we found it important to not exclude higher BMI groups from the intervention. As a feasibility intervention we are interested to find out what types of people access the intervention, engage with it and what are their experiences and feedback. This will give us meaningful information for designing a future trial.

Feedback 11: Given the acknowledged difficulties in recruiting men would it not be advisable to target men via all strategies?

Author response 11: This was an issue with our chosen wording. We have amended the wording (see p9) to clarify that all strategies targetted both men and women, but in particular, our local knowledge enabled us to target males on a local level e.g. local football stadiums etc.

Feedback 12: Provide details of the recruitment pack via the SPCRN, specifically details on how the study is introduced to these potential participants and how they were identified.

Author response 12: We have included the contents of the recruitment pack as requested (see p9). Details of how participants were identified (searching the patient databases based on the study inclusion and exclusion criteria) are already included in the original paragraph.

Feedback 13: Will detail of the number of participants who opt for either home assessments or the research lab assessments be reported on?

Author reponse 13: We will keep a log of where participants choose to be seen and report the findings. This has now been added to Table 4 (p24).

Feedback 14: Are incentives provided to participants for completing assessments?

Author response 14: Participants receive £20 shopping voucher for each data collection time point (baseline, follow-up and optional interviews). This is already described in the manuscript (under 'Retention' p33).

Feedback 15: Is it possible that for some individuals internet access is by smartphone only. Therefore is the website constructed using a responsive design approach which will be viewable on such a device?

Author response 15: Yes. This was an important element of our intervention development phase. The software company worked on developing a website that could be accessed via smartphones. This has now been clarified within the text (see p11).

Feedback 16: P1 I32 – given that the app is used to set goals and self-monitor would it not be useful for people to access this information via the app?

Author response 16: The participants are able to set goals and self-monitor via the app. It is only the advice and further information on goal setting etc that is on the website (see Table 1). During our intervention development phase, insight from participants and experts from the software company highlighted the need for the app to focus on several key features. Substantial amounts of text or content on apps were highlighted as barriers to engagement. Participants from our development phase highlighted the preference for a separate website which contained more detailed information. Our daily messages to the app are based on website content, so participants do receive snippets of useful evidence based information every time they open the app.

Feedback 17: Is the helper reinforcement limited to sending 'smiles?' Can they provide text based support in the form of comments?

Author response 17: Helpers and participants can interact in several ways. This includes (i) sending animated smiles, some with and without preset messages; (ii) sending text based messages via their preferred platform (e.g. SMS, WhatsApp etc.); (iii) making phone calls; and (iv) meeting and talking face-to-face. Messaging and phone calls have quick access entry from the HelpMeDolt! App. This has now been clarified in the manuscript (p11-17).

Feedback 18: How is motivational language operationalised in the app?

Author response 18: The website provides examples of motivational language that can be used by helpers to support their friend. This motivation can be provided via phone call, in person, via text message or by sending 'smiles' via the app. A range of smiles include simple motivational language, such as, "Keep up the good work!" or "Great job this week". Some examples are now included in the manuscript (see p15).

Feedback 19: Description of the helper intervention content is necessary.

Author response 19: Additional detail has been added to the 'intervention' methods section (see p11-17).

Feedback 20: P12 App content. More detail is necessary on how these aspects are operationalised and implemented in the intervention. What other methods of interacting are available? Is there any opportunity for two way live communication such as messaging

Author response 20: Additional detail has been added to the text. We have also included Table 1 which outlines how all elements of the logic model are operationalised within the app and website (p11-17).

Feedback 21: More detail is necessary on the methods used to prompt engagement.

Author response 21: Additional detail has been added as requested (p16).

Feedback 22: Is there any geographic restriction to where helpers can be located?

Author response 22: No. Helpers can be based anywhere in the world. This has been clarified in the manuscript (see p16).

Feedback 23: Inclusion criteria is people trying to lose weight and there are no exclusion criteria regarding recent weight loss. How will the trial deal with people who have already lost a significant amount of weight or current use another weight loss strategy.

Author response 23: This is a pragmatic trial and is therefore inclusive of a wide range of people trying to manage their weight. People who have recently lost weight, but continue to have a BMI>30 may still wish to lose more weight, or sustain the weight they have already lost. Social support can be an integral part of this process. The nature of the feasibility study is to explore the characteristics of participants who sign up to a technology based intervention. This helps us understand the context in which an intervention like this would be used within the general population, where people may often use an app in addition to other weight loss techniques This app is designed to be used alone or along with other weight loss approaches. In a future powered trial the randomisation should also ensure that participants who have recently lost weight and/or use another weight loss programme would be randomised evenly between the intervention and control group.

Feedback 24: Please justify the choice of 15 sec epoch

Author response 24: We have added additional information related to the accelerometer analysis (p21-22). This includes clarification that our data was collected in 1sec epochs and analysed as 15sec epochs. This has been guided by previous research which shows that suggests using longer epochs (e.g. >30sec) may reduce the amount of physical activity detected.

Feedback 25: Details should be presented on the methods used to process the data.

Author response 25: Thank you for sending the Montoye reference. We have adhered to their recommendations for reporting and now include additional detail as requested (p21-22).

Feedback 26: What access to the app will helpers have?

Author response 26: Helpers will be able to see all the data entered by their friend, including: weight and weight progress, goals and goal progress, and number of smiles received. This extra detail has now been added to the manuscript (p16).

Feedback 27: Provide clarification around the sample size justification. The main outcomes are BMI, physical activity and diet, which are presumably continuous measures so do the authors mean they are powered to detect an 11 point percentage difference in these scores?

Author response 27: As a feasibility study, we will not be conducting formal hypothesis testing of the putative primary outcomes for the future effectiveness trial (in line with feasibility reporting guidelines (Reference: <http://www.bmj.com/content/355/bmj.i5239>).

As a feasibility study the primary outcomes are instead feasibility outcomes such as the proportion of participants retained, the proportion of participants who found being involved in the study acceptable, the proportion of participants who provided analysable outcome data. Our sample will provide enough precision to estimate these proportions across the whole sample to within plus or minus 11 percentage points using a 95% confidence interval. We have clarified this in the text (p31).

Feedback 28: As both accelerometer and self-report are collected for physical activity how is the decision made regarding if the intervention successfully change PA if both these measures are not significant?

Author response 28: Our current methods are primarily exploring the feasibility of using different physical activity measures for a future effectiveness trial. The feasibility study is not powered to detect significant changes in outcomes.

Feedback 29: Consider expanding the analysis of usage data to examine patterns and trends over time.

Author response 29: We have now added detail to clarify this in the manuscript (see Table 4).

REVIEWER 2

Feedback 30: The introduction should be revised in order to more clearly focus on existing literature regarding mHealth and weight loss trials and how this trial fits there.

Author response 30: The requested changes have been made to the introduction (p6).

Feedback 31: Please revise so that the aim in the abstract and the aim in the end of the introduction are more consistent. The aim in the introduction does not mention weight loss.

Author response 31: The aims have been clarified in the Introduction (p7).

Feedback 32: Why targeting the wide age range of 18-70 years? A 18-year-old may have very different needs than a 70-year-old in regards to a weight loss app intervention.

Author response 32: The intervention has been designed to be useable for the 18-70 age group. The weight loss strategies used are accessible and relevant to all age groups and the intervention is able to be tailored to individuals, e.g. they choose their own goals and how to interact with helpers etc. Our stage 1 development phase ensured we collected insights from a wide age range of users, including 65+yrs, so that their age specific input was included in development.

Feedback 33: Why keep women that become pregnant in the study? Weight data cannot be used and also other variables may be difficult to assess.

Author response 33: Weight gain can often be an issue during pregnancy, in addition to weight retention in the postpartum, especially for women who already have a BMI>30. Following a healthy diet and engaging in safe physical activity is recommended. The social support offered by this intervention may be an important tool in supporting women with healthy lifestyle through pregnancy and into the postpartum. Removing their access to the intervention after becoming pregnant may have a detrimental effect. Anyone who becomes pregnant during the study is provided with appropriate evidence based guidance on healthy eating and physical activity while pregnant. In a future powered trial the randomisation should also ensure that pregnant women are distributed evenly between arms and therefore would not affect outcomes.

Feedback 34: What is the rationale for doing a combined intervention with a website and an app? Why not do only an app since they today can contain a lot of information and it would make it easier with only one medium for the participants?

Author response 34: This point was also raised by the other reviewer. Please see the author response for feedback no.16.

Feedback 35: The authors should describe and motivate the dietary outcomes that will be assessed in detail.

Author response 35: Additional detail has been added (see p22).

Feedback 36: Why not standardise weight measurements to include no clothing and preferably before breakfast?

Author response 36: In a pragmatic trial this is difficult to achieve, participant appointments are made at a time and location convenient for participants meaning that measuring weight before breakfast is typically not feasible. We were keen to remove barriers to participation and would not like to make participants with obesity feel uncomfortable by asking them to remove their clothes for measurement.

Feedback 37: The authors should provide some more details on the Actigraph measurements e.g. sampling frequency, and filter to be used for analysis. And why choose the waist? Have you considered wrist-worn?

Author response 37: Additional detail has now been added to the manuscript regarding our Actigraph protocol (p22-23). We explored wrist worn Actigraphs but found the existing evidence for waist worn devices to be stronger at the time of our protocol.

Feedback 38: Why two methods for dietary intake? Have the methods been validated in obese subjects? How will portion sizes be assessed?

Author response 38: Two methods are being assessed and compared for feasibility and acceptability for use in the future trial Additional detail in relation to these measures has been added to the manuscript (p23).

Feedback 39: Should the study protocol adhere to the CONSORT-EHEALTH: improving and standardizing evaluation reports of Web-based and mobile health interventions. J Med Internet Res 2011;13(4):e126?

Author response 39: In line with guidance from BMJopen the protocol manuscript has adhered to the SPIRIT guidelines (included in the appendices). We acknowledge your suggestion of the CONSORT-eHealth checklist and will strive to achieve each of the checklist items when reporting the study results.

Feedback 40: Minor: the authors' contribution paragraph is very long.

Author response 40: The acknowledgements section has been written in line with the guidance provided by BMJopen and remains unchanged.

VERSION 2 – REVIEW

REVIEWER	Marie Lof Karolinska Institutet, Sweden
REVIEW RETURNED	05-Jul-2017

GENERAL COMMENTS	The authors have responded very well to my previous comments.
---